# Efficacy of Edible Bird’s Nest on Cognitive Functions in Experimental Animal Models: A Systematic Review

**DOI:** 10.3390/nu13031028

**Published:** 2021-03-23

**Authors:** Maznah Ismail, Abdulsamad Alsalahi, Musheer Abdulwahid Aljaberi, Ramlah Mohamad Ibrahim, Faizah Abu Bakar, Aini Ideris

**Affiliations:** 1Laboratory of Natural Medicines and Products, Institute of Bioscience, Universiti Putra Malaysia, Serdang 43400 UPM, Selangor, Malaysia; ramlah86ibrahim@gmail.com; 2Department of Pharmacology, Faculty of Pharmacy, Sana’a University, Mazbah District, Sana’a 1247, Yemen; ahmedsamad28@yahoo.com; 3Department of Community Health, Faculty of Medicine & Health Sciences, Universiti Putra Malaysia, Serdang 43400 UPM, Selangor, Malaysia; musheer.jaberi@gmail.com; 4UZMA Clinic, Kelana Mall, SS6/12 Road, SS 6, Petaling Jaya 47301, Selangor, Malaysia; dr.faiza.bakar@gmail.com; 5Department of Veterinary Clinical Studies, Faculty of Veterinary Medicine, Universiti Putra Malaysia, Serdang 43400 UPM, Selangor, Malaysia; aiini@upm.edu.my

**Keywords:** cognition, neurodegeneration, neuroprotection, edible birds’ nest, systematic review

## Abstract

Edible bird’s nest (EBN) is constructed from saliva of swiftlets birds and consumed largely by Southeast and East Asians for its nutritional value and anti-aging properties. Although the neuroprotection of EBN in animals has been reported, there has not been yet systemically summarized. Thus, this review systemically outlined the evidence of the neuroprotective activity of EBN in modulating the cognitive functions of either healthy or with induced-cognitive dysfunction animals as compared to placebos. The related records from 2010 to 2020 were retrieved from PubMed, Scopus, Web of Science and ScienceDirect using pre-specified keywords. The relevant records to the effect of EBN on cognition were selected according to the eligibility criteria and these studies underwent appraisal for the risk of bias. EBN improved the cognitive functions of induced-cognitive dysfunction and enhanced the cognitive performance of healthy animals as well as attenuated the neuroinflammations and neuro-oxidative stress in the hippocampus of these animals. Malaysian EBN could improve the cognitive functions of experimental animals as a treatment in induced cognitive dysfunction, a nutritional cognitive-enhancing agent in offspring and a prophylactic conservative effect on cognition against exposure to subsequent noxious cerebral accidents in a dose-depended manner through attenuating neuroinflammation and neuro-oxidative stress. This systemic review did not proceed meta-analysis.

## 1. Introduction

Several interrelated cerebral pathways establish the etiopathogenesis of neurodegeneration such as neuroinflammation, altered gene expression, and oxidative stress that eventually could result in the death of neuronal cells [1,2], which are manifested clinically as cognitive dysfunction [3]. Even if there are available medicines that are used for Alzheimer’s disease, there is a need for more therapies since these medicines do not remedy the whole spectrum of pathobiology [4,5].

Edible bird’s nest (EBN), which is constructed by Swiftlets birds (Collocalia species) from their saliva [6,7,8,9,10], is consumed by Asians especially Chinese for its nutritional value and some other medicinal benefits as claimed in their folk medicine of Southeast and East Asian countries [11,12,13,14,15,16,17]. Chemically, EBN is rich in carbohydrates including sialic acid, galactose, glucosamine, galactosamine and N-acetylneuraminic acid [17,18]. Sialic acid has been reported to have an important role in the development of brain of mammals and intelligence, which could improve learning and memory abilities through long-term strengthening potentiation in hippocampus [18]. In addition, EBN contains lactoferrin and ovotransferrin (glycoproteins as members of transferrin), which were reported to contribute in the neuroprotective activity through scavenging free radical species in SH-SY5Y cells [19].

As a matter of fact, the literature is rich in studies that have reported the pharmacological effects of EBN, including those related to neuroprotection [18,20,21,22,23,24,25,26], and several reviews have outlined the pharmacological properties of EBN in several health conditions [27,28,29,30]. To the best of our knowledge, the literature lacks a systematic review on the neuroprotective activity of EBN. Thus, this study reviewed the literature to systemically outline the evidence of the neuroprotective activity of EBN in modulating the cognitive functions of animals with induced- or transgenic cognitive dysfunction as compared to a concomitant appropriate placebo in animal-based studies.

## 2. Materials and Methods

The Preferred Reporting Items for Systematic Reviews and Meta-Analysis (PRISMA) [31,32] were followed strictly during the whole stage of this systematic review (design, conduct and reporting). The prospective protocol of this systematic review was registered at PROSPERO platform [ID: CRD42021225056].

### 2.1. Search Strategy

The constructed keywords were Edible bird’s nest AND Nervous System; Edible bird’s nest AND Neurodegenerative Diseases; Edible Birds’ nest AND Brain; Edible birds’ nest AND Cognition. The online databases of PubMed, Scopus, Web of Science and ScienceDirect were the source of the related records. Moreover, the lists of references of the published articles and Google Scholar were searched to ensure comprehensive retrieval of similar works on the effect of EBN on the cognitive functions in the animal models. The time frame of retrieving the pertinent records was from 2010 to 12th of December 2020 without refining the languages, countries, types of manuscripts or study design. For avoiding publication bias, ProQuest and TRIP Database were searched to retrieve unpublished thesis or proceeding papers. Research strategy was implemented by two independent investigators (A.A. and M.A.A.), while a discussion with a third investigator (M.I) was consulted in case of none consent.

### 2.2. Study Selection

Duplicates were removed, followed by removing the secondary literature (e.g., books, book sections, reviews) and other documents (e.g., protocols, guidelines, lectures and conference abstracts) for restricting selection to the primary literature (original research articles). Then the remaining records were screened by titles and abstracts to remove unrelated articles to EBN. Through full text screening, the remaining records subjected to secondary selection according to the prespecified eligibility criteria (prepared according to the PICOS model) to include animal-based studies that evaluated the effects of EBN on cognition functions, (Table 1). The eligible studies were further subjected to appraisal of internal validity (assessment of the risks of bias). Study selection was implemented by two independent investigators (A.A. and M.A.A.), while a discussion with a third investigator (M.I) was consulted in case of none consent.

### 2.3. Assessment of the Risk of Bias

The assessment criteria for each domain of bias in these included animal-based studies was adopted according to SYRCLE’s tool [33]. The assessment of the risk of bias was performed at the level of the studies involving the evaluation of selection bias, performance bias, attrition bias, detection bias, reporting bias and other sources of bias. The answer of the signaling questions were either a low, high or unclear risk of bias to indicate the presence, absence or uncertain level of bias, respectively. Then the risk of bias was assessed within each included study and across the whole studies. After that, any study showed more than four domains of high risks, it was excluded [34]. The appraisal of the risks of bias was implemented by two independent investigators (A.A. and M.A.A.), while a discussion with a third investigator (M.I) was consulted in case of none consent. The selection bias was evaluated through addressing for random allocation, similar baseline characteristics and allocation concealment. The performance base was evaluated through adequate addressing for random housing of animals’ cages and blinded administration of the intervention. For detection bias, it was evaluated through the assessment of random selection of animals to measure the outcomes, and the blinded measurement of outcomes. For attrition bias, it was evaluated through the evidence of adequate addressing of the missing of data and animals. For reporting bias, it was evaluated through focusing on whether the primary and secondary outcomes were reported transparently and adequately. For other source of bias, it was evaluated through looking of funder bias in favor of a beneficiary company.

### 2.4. Data Extraction and Collection

The data were collected from tables and texts of the manuscripts. Such data were extracted and filled in a spreadsheet of excel (table of study characteristics). If there were missed data, authors were contacted through emails. The data were extracted by two independent investigators (A.A. and M.A.A.) and a discussion with a third investigator (M.I) in case of discrepancy.

Study design data were study ID (first author and year), type of design (parallel, crossover); disease modeled (healthy, transgenic diseased, induced-diseased) and settings (location and country where the study conducted).

Population data were animal species (rats, mice, rabbits or primates), strains, sex (male or female), age (pub, young, adult, elderly), number of total animals (N) and genetic background of animals (normal or transgenic).

Intervention data were the name of intervention, dose levels (mono or multiple levels in weight/weight or weight per volume), frequency of dosage (once or twice per a day), intervention duration (days, weeks, months…etc.), vehicle in which the intervention formulated (vehicle name, active or inert, volume), route of administration, number of animals in the intervention group.

Comparator data were nature of the comparator, vehicle, frequency of dose (once, twice…etc.), route of administration, number of animals in the comparator group.

Primary outcome data were spatial learning and memory, [measured by escape latency defined as the average time to find the hidden platform in seconds using Morris Water Maze Behavioral Test or time spent in the novel arm (in seconds) of Y-maze test].

Secondary outcomes data were neuroinflammation [measured by levels of TNF-α (pg/mL) in hippocampal tissue supernatant]; apoptosis [measured by the level of caspa-se-3 (pg/mL) in hippocampal supernatant]; oxidative stress [antioxidant activity measured by the levels of superoxide dismutase (U/L) in hippocampal supernatant and fold changes in the level of reactive oxygen species in hippocampal supernatant, oxidant activity measured by the level of malondialdehyde (nmol/L) in hippocampal supernatant].

### 2.5. Data Synthesis and Reporting

This systematic review was planned to conduct a narrative synthesis for the extracted data by summarizing changes in the primary or secondary outcomes to enable making a decision about the direction of evidence of the efficacy of EBN on cognitive functions and hippocampal neuroprotection against neuroinflammation, apoptosis and oxidative stress. In association with summarizing the outcomes, the data of dose level, route of administration, and duration of intervention were included to enable evaluating the consistency of the evidence of the efficacy of EBN on cognition and the neuroprotection of EBN against neuroinflammation, apoptosis and oxidative stress, and how deviations could affect the observed outcomes across the included studies.

## 3. Results

### 3.1. Sulected Studies

A total of 355 records were retrieved form the four online databases and one record from extra sources. According to the eligibility criteria, six relevant studies to the effect of EBN on cognitive functions were included [18,20,21,22,25,26], (Figure 1).

### 3.2. Appriasal of the Risk of Bias (Internal Validty)

The risk of bias across the six included studies was observed in the blinding of the assessors during selection of animals, preforming the interventions and detection of the outcomes, while the attrition bias, reporting bias and other sources of bias were adequately practiced, (Figure 2A,B).

### 3.3. Study Characteristics

A total of six animal-based studies were included in this systematic review [18,20,21,22,25,26] covering a timeframe of 7 years from 2014 to 2020 and conducted in several educational institution in Malaysia (*n* = 5) [20,21,22,25,26] and China (*n* = 1) [18]. All the included studies were parallel interventional studies aimed to investigate the effect of EBN on cognitive functions of experimental animals as a primary outcome [18,20,21,22,26], except for the study by Zhiping et al. [25] aimed to investigate the effect of EBN on oxidative stress in the hippocampal of animals. However, the health status of the recruited animal models was different, which measure the effect of EBN in animals with normal cognitive functions [18,26], ovariectomized-induced cognitive dysfunction [21,22,26] or lipopolysaccharide-induced cognitive dysfunction [20]. In accords to the latter aspects, the studies provided EBN to animals either for treating the induced-cognitive dysfunctions [21,22,25], investigating how nutritional EBN could enhance cognitive functions of offspring [18,26], or investigating the prophylactic effect of EBN on cognitive functions prior to inducing cognitive dysfunctions [20].

All the included studies investigated the effect of EBN against the effect of placebos, of which vehicle, population characteristics and baseline characteristics were identical to those of the intervention. Additionally, the EBN interventions in the included studies were collected from several states and regions of Malaysia. However, different routes were used to deliver EBN to animals either via oral via oral gavages [20], mixed with rodent diets [21,22,25] or allowing pups to suckle milks from mothers receiving oral EBN [18,26], Similarly, the durations of intervention in the included studies were different and took a range from 7 days to 142 days.

For populations, the recruited animals in the included studies were rodents, which were either ICR mice [18], CJ57B/6 mice [26], Sprague-Dawley rats [21,22,25] or Wister rats [20]. The total sample size of animals in the included studies was 104, which were divided equally between the interventions and controls. Regarding to sex of animals, mostly female animals were the most predominant population [21,22,25], except for the study by Careena et al. [20] that recruited male animals, while the studies by Xie et al. [18] and Mahaq et al. [26] recruited male and female animals. Moreover, three studies recruited elderly animals (3 months or 12–14 weeks of age), [21,21,22,25], while two studies recruited offspring cubs [18,26], (Appendix A).

### 3.4. Outcomes

#### 3.4.1. Chemical Analysis of EBN

The included studies in this systematic review focused on the effects of EBN on cognition in experimental animal models. However, the content of 5 mg of Malaysian EBN from different sources (East Malaysia, Northern region, West coast, Southern region, heavily polluted industrial area, and East coast) was analyzed by using high performance liquid chromatography with fluorescence detection. The results showed that the different percentages of the content of sialic acid in the samples of EBN taking a range from 5.47 to 11% [20]. Similarly, the study by Mahaq et al. [26]. identified the sialic acid content of the used samples of naturally collected EBNs from North Malaysia, South Malaysia and Brono Sabah by standardized high performance liquid chromatography. The results indicated that each 0.01015 g of EBN contain sialic acid in different percentages of 2.97 ± 1.63 (North Malaysia), 3.15 ± 0.34 (South Malaysia) and 2.02 ± 1.76 (Brono Sabah) [26].

#### 3.4.2. Spatial Learning and Memory (Cognition)

Allowing female elderly rats with ovariectomy-induced cognitive functions *to ad libitum* feed on 1.2, 0.6 or 0.3 g/kg EBN for 90 [22] or 140 days [21] resulted in a dose-dependent improvement of the spatial learning and memory as evidenced by the significant shorter escape latency. Similarly, oral feeding with a once daily dose of 125, 250 or 500 mg/kg of EBN for 7 days prior to Lipopolysaccharides-induced cognitive dysfunction in elderly male Wistar rats resulted in a dose-dependent prevention of further deterioration of cognitive functions through a significant conservation of higher spatial learning and memory as evidenced by the significant shorter escape latency [20]. Moreover, allowing newly born offspring mice (first generation) to suckle maternal milk of mother mice exposed to oral dose of 4.5, 6.75 or 9 g of EBN for 42 days resulted in enhancing cognition through improving the spatial learning and memory as evidenced by the significant longer staying in the third quadrant of Y-maze [18], however, the 9 g of EBN was the only dose that exerted a significant improving of spatial learning and memory [18]. Similarly, allowing newly born offspring mice (first and second generations) to suckle maternal milk of mother mice exposed to oral dose of once daily 10 mg of different natural sources EBN (North Malaya, South Malaysia, and Brono Sabah) of EBN for 42 days resulted in improving the spatial learning and memory as evidenced by the significant longer staying in the third quadrant of Y-maze [26].

#### 3.4.3. Neuro-Inflammation and Neuro-Oxidative Stress

Although allowing elderly female rats with ovariectomy-induced cognitive dysfunction to *ad libitum feed on 60*, 30 or 15 g of EBN exerted a dose-dependent attenuation of hippocampal oxidative stress as evidenced by the levels of superoxide dismutase and catalase, the levels of superoxide dismutase and catalase in all EBN doses remained significantly lower than those of control. However, the hippocampal levels of MDA were significantly lower with all the EBN doses as compared to the control [25]. Nonetheless, allowing newly born offspring mice (first and second generations) to suckle maternal milk of mother mice exposed to oral dose of once daily 10 mg of different natural sources EBN (North Malaysia, South Malaysia, and Brono Sabah) of EBN for 42 days resulted in a significant attenuation of hippocampal oxidative stress as evidenced by the high levels of superoxide dismutase enzyme and the lower levels of malondialdehyde [26]. Similarly, oral feeding with a once daily dose of 125, 250 or 500 mg/kg of EBN for 7 days prior to Lipopolysaccharides-induced cognitive dysfunction in elderly male Wistar rats resulted in a significant dose-dependent attenuation of hippocampal neuroinflammation as evidence by the reduced level of Reduce TNF-α, IL-1β, and IL-6 as well as significant dose dependent attenuation of the hippocampal oxidative stress as evidenced by the lower levels of reactive oxygen species and malondialdehyde [20].

#### 3.4.4. Underlying Molecular Mechanism

The evidence from the preclinical studies showed that the cognitive enhancing activity of EBN on hippocampal neurons (human SH-SY5Y neuroblastoma cells) [19,35] could be due to increasing the SIRT1 expression in the pyramidal layer and dentate gyrus of the hippocampus [21]. Additionally, EBN could provide a neuroprotective activity through its antiapoptotic activity that was evidence by inhibiting early apoptotic membrane phosphatidylserine externalization as well as inhibition of caspase-3 cleavage. in SH-SY5Y cells [35]. Moreover, EBN could provide a neuroprotective activity by exerting an antioxidant effect that protect the hippocampal cells against oxidative stress through attenuating ROS build up [19,35] and increase up-regulating SOD genes such as SOD1 and SOD2 in human SH-SY5Y cells [36], (Figure 3).

## 4. Discussion

It is known that sialic acid is the main bioactive constituent of EBN [37], and it was reported that it is responsible for the neuroprotective effect of EBN. Sialic acid was identified in the EBNs from several regions of Malaysia and its percentage was slightly different from one EBN to another [20,26]. In addition, EBN contains lactoferrin and ovotransferrin (glycoproteins as members of transferrin), which were reported to contribute in the neuroprotective activity through scavenging free radical species in SH-SY5Y cells [19]. However, the effects of EBN may not be explained by the presence of a single bioactive in view of the wide variety of its components. Accordingly, the multiple bioactive compounds in EBN may have produced their effects through synergism.

The findings of the risks of bias assessment indicated that blinding of the assessors during selection of animals, preforming the interventions and detection of the outcomes were not adequately implemented, while the randomization of animals, attrition bias, reporting bias and other sources of bias were adequately practiced. The former drawbacks are highly common in animals-based studies because the allocation of animals, intervention administration, measurement of outcomes and analysis of the data are usually conducted by the same assessor, which means that blinding of the assessor would unavoidable [33]. Additionally, randomization of animals has not been yet a standard practice like the randomized controlled trials, and the heterogeneity among animals is not assumed like humans since the recruited animals share identical genetic and health characteristics [33]. Even if the whole assessed domains of bias were of low risk of bias, the internal validity of animal-based studies are general relatively lower than that of the randomized controlled trials [33]. Undoubtly, the detection bias was not adequately addressed, which could lead to an overestimation of the effect size of the neuroprotective activity of EBN because of the authors preference during the outcome measurement. However, the detection bias across the included studies and within each study was unclear because the absence of direct and indirect evidence to be well judged. Although there is positive evidence of the neuroprotective activity of EBN in this systematic review, the generalizability of this evidence has been still controversial since evidence form animal-based studies could not always be extrapolated to human, but the evidence of this systematic review could be valuable for future research towards conducting clinical trials for evaluating the safety and efficacy of the neuroprotective activity of EBN, particularly that the EBN has a long history of safe consumption by the population of South Eastern countries.

Up to now, there are no clinical trials that have reported the efficacy of the neuroprotective activity of EBN, which may give rise to the need for conducting randomized controlled trials.

The primary objective of the 6 included studies was to evaluate the effects of EBN on cognitive functions of animals with normal cognitive functions or induced-cognitive dysfunction. Moreover, the primary outcome in all the included studies was spatial learning and memory using standard reliable behavioral tests (e.g., water maze and Y-maze) [36]. The former findings supported the consistency and reliability of the evidence the neuroprotective effect of EBN on cognition across the included studies.

Combining the studies’ characteristics indicated that this systematic review addressed several aspects of the neuroprotective effects of EBN through evaluating such outcome in the purpose of either treating induced-cognitive dysfunction, providing prophylactic neuroprotection prior to exposure to a neurodegenerative insult or as enhancing the cognitive functions in offspring during lactation. Moreover, the neuroprotective effects of EBN were elucidated in terms of applying different routes of administration of EBN (oral gavage, mixed with diet or suckling maternal milk of EBN-treated mothers), different treatment courses (short- or long-term administration from 7 days to 142 days) and different sources of natural EBN. Furthermore, the six animal-based studies that were included in this systematic review covered a timeframe of 7 years from 2014 to 2020 and conducted in several educational institution in Malaysia, an aspect reflecting the interest in nutritional EBN-based industry and consumption of EBN by Malaysian people [38]. Finally, the findings indicated that the included studies recruited male and females, however, female animals were the most predominant population, which an aspect adding a particular importance to the findings of this systematic review since most behavioral studies avoid females animals due to that estrous cycle of female animals could result in producing unreliable data [36]. Consequently, the study characteristics could indicate that the included studies considered the effects of various factors that could affect the neuroprotective effects of EBN on cognitive functions.

Whatever the source of EBN, route of administration, duration of treatment and purpose of therapy (nutrition, treatment or prophylaxis), the findings of this systematic review indicated that the Malaysian EBN could improve the cognitive functions of experimental animals as a treatment in induced-cognitive dysfunction, a nutritional cognitive-enhancing agent in offspring and a prophylactic conservative effect on cognition of animals against exposure to subsequent noxious cerebral accidents, particularly that antioxidant activity of Peninsular Malaysia EBNs were approximately two times greater than the other EBNs [37]. Perhaps the neuroprotective effects of EBN on cognition could be due to the consent of sialic acid as an the main bioactive component of EBN [37]. The efficacy of EBN was not time-dependent since EBN exhibited a neuroprotective effect in animal models with different health status, ages and sex over short- or long-term interventional schedules, however, the neuroprotective effect of EBN was dose-dependent. Perhaps the reported cognitive enhancing effect of EBN as evidenced by the findings of this systematic review could be attributed to its ability to attenuate the neuroinflammations and oxidative stress. 

Unfortunately, the included studies in this systematic review did not report the findings of the spatial learning and memory were being graphically presented which makes the extraction of value of measurements of the primary and secondary outcomes as mean (standard deviation) inaccessible. Accordingly, this systematic review could not proceed meta-analysis to validate the evidence of the qualitative systematic review. Since the evidence of the neuroprotective effects of EBN on cognitive functions were pooled from preclinical animal-based studies, this systematic review could not be extrapolated to be applicable in humans. However, the pooled evidence of EBN neuroprotective effects of EBN provides interesting indicators to conduct clinical observational trials, quasi interventional studies or even randomized controlled trials on the neuroprotective effects of EBN on cognition, particularly EBN is consumed by Asian people for decades [39].

## 5. Conclusions

Malaysian EBN could improve the cognitive functions of experimental animals as a treatment in induced cognitive dysfunction, a nutritional cognitive-enhancing agent in offspring and a prophylactic conservative effect on cognition of animals against exposure to subsequent noxious cerebral accidents. Additionally, the efficacy of the neuroprotective effects of EBN was dose-dependent, which could be attributed to ability of EBN to attenuate the neuroinflammations and neuro-oxidative stress.

## Figures and Tables

**Figure 1 nutrients-13-01028-f001:**
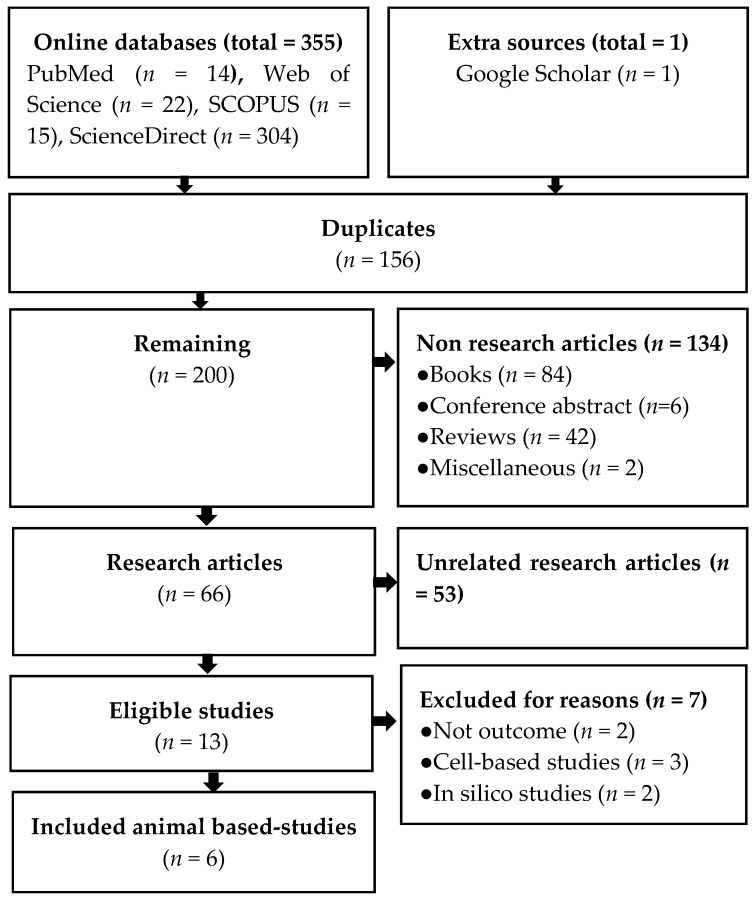
PRISMA flow diagram.

**Figure 2 nutrients-13-01028-f002:**
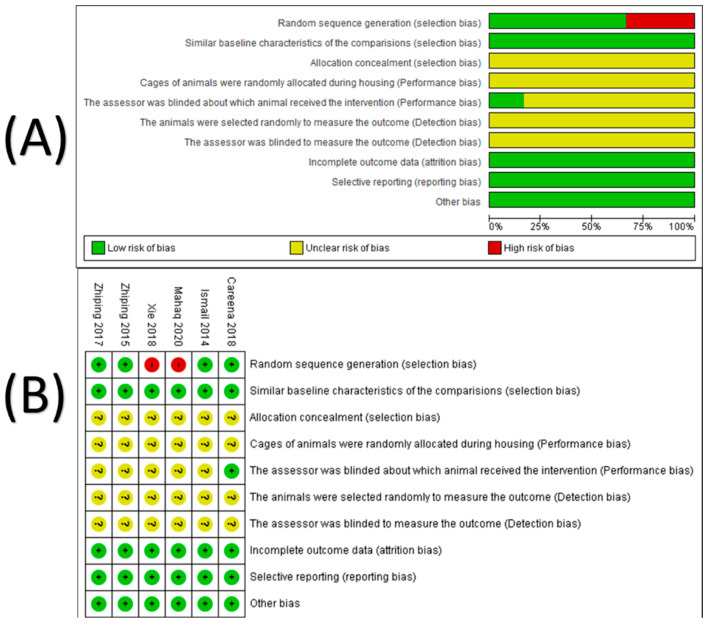
Assessment of risk of bias; (**A**): risk of bias across the included studies, (**B**): risk of bias within each study. Green color indicated a low risk of bias, Yellow color indicated unclear risk of bias, red color indicated high risk of bias. Careena et al., 2018: [20], Ismail 2014: [22], Mahaq 2020: [26], Xie 2018: [18], Zhiping 2015: [25], Zhiping 2017: [21].

**Figure 3 nutrients-13-01028-f003:**
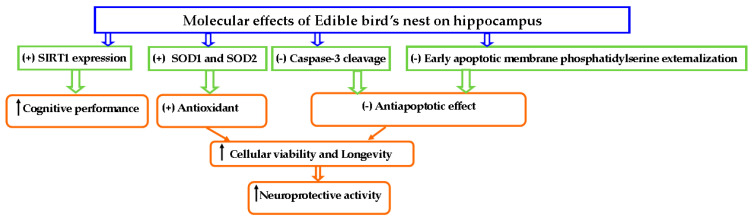
The molecular mechanism underlying the neuroprotective effects of EBN. (+): stimulation or upregulation, (-): inhibition or attenuation. SOD1 and SOD2: genes of superoxide dismutase, Upward arrow: enhance.

**Table 1 nutrients-13-01028-t001:** Eligibility criteria (inclusion and exclusion criteria).

Items	Included	Excluded
Human disease model of animals	Cognitive functions in healthy, surgically-induced cognitive dysfunctions, chemically-induced cognitive dysfunctions, nutritionally-induced cognitive dysfunctions or transgenic-induced cognitive dysfunctions	Anxiety, multiple sclerosis, epilepsy, parkinsonism
Population	Animals at any age stage (pups, young, young adults, adults or elderly); any gender (males and females); any species (rats, mice, rabbits, primates), any strains of those species	Cells and humans
Intervention	Edible birds’ nest (crude, any dose, any timing, ant dose frequency, number of animals received the intervention)	Isolated components of the edible birds’ nest Edible birds’ nest combined with other supplements or medicines
Comparator	Concomitant negative control untreated with EBN with similar baseline characteristics to that of intervention, exposed to similar treatment conditions to that of the intervention, and similar animal species to that used in the intervention group	Positive controls
Study design	Preclinical interventional controlled animal-based-studies with any design (parallel or crossover), acute, sub-acute or chronic exposure to mono-level or multi-level dosing of EBN with a concomitant and appropriate control, implemented anywhere, conducted from 2010 to 2020.	In silico-based studies In vitro cells-based studies Human studies
Primary outcome	Cognition in terms of spatial learning and memory	Locomotive activity, anti-anxiety or anti-epilepsy
Secondary outcome	Hippocampal inflammation in terms of levels of inflammatory markers Hippocampal oxidative stress in terms of levels of antioxidant and oxidants markers Hippocampal apoptosis in terms of levels of caspases	Inflammation, oxidative and apoptosis in serum or organs other than hippocampus
Article type	Published and unpublished research articles Full papers in proceedings Unpublished theses Manuscripts written in any language	Published theses Inaccessible research articles

## Data Availability

Data are included in manuscript.

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
