# Peer review of "Efficacy of Edible Bird’s Nest on Cognitive Functions in Experimental Animal Models: A Systematic Review"

_nutrients, 2021, doi:10.3390/nu13031028_

Round 1
Reviewer 1 Report
The review of Maznah Ismail et al summarizes and analyses the literature of edible Bird´s Nest (EBN) on neurodegeneration and cognitive function in animal models.
Several reviews about the positive properties of EBN in human health exist. However, this review focuses on the positive aspects of EBN in respect to neurodegeneration and cognition in animal models, which has not been published in this context before. Moreover, I would like to emphasize that search strategy, study selection assessment of the risk of bias as well as data extraction is precisely described and transparently displayed, which is in many reviews neglected. In line, the analysis of the mainly 6 studies are reported balanced and gives a comprehensive and contemporary overview of this topic. In my opinion the review is in the clear scope of the journal and interesting for a broad readership, especially for scientists dealing with the impact of nutrition/nutrients in neurodegeneration and brain function. As far as I am concerned, I have only some mainly minor suggestions before this review is suitable for publication:
Principal concerns:
- The authors restrict themselves on animal studies. This is already clearly pointed out in the title of the review. However, especially some cell culture experiments reveal the underlying molecular mechanisms of EBN in neuroprotection. I would appreciate if, although this is not the main focus of this review, the authors would pay a little bit more attention on the molecular mechanisms in general and include these cell culture experiments.
Minor concerns and suggestions:
- A table summarizing the articles reviewed would help to get a quick overview of this topics (and in my experience increases the citation of the review)
- Same for a graphic summarizing the underlying (molecular) mechanisms. I think this would further increase the impact of the paper
- line 43. I would prefer another wording. For Alzheimer´s disease which is the most prominent or common neurodegenerative disease, no causative treatment is available
- The introduction is really short. In particular information about the potential bioactive compounds are missing.
- At least in the discussion or conclusion a short paragraph about clinical studies (if available) would be interesting.
(- page 5 empty)
Author Response
The response has been attached

Reviewer 2 Report
In the current article, authors summarized the evidence provided from animal-based studies investigating the neuroprotective activity of EBN in cognition. The topic of the review is very interesting and the manuscript is well written. Authors provide essential information and synthesize the existing literature. They have organized their findings based on different criteria and critiqually summarize the strengths and limitations of the current literature.
I have some very minor suggestions. First of all, lines 98 to 113, where the bias are described, would be better if slightly rephrased. A rephrase would make clearer the way bias were evaluated.
Moreove, although authors discuss about the bias of the existing studies (lines 243-250) regarding the assessors blinding and preformance of the interventions, they could further discuss on the consequences of these bias in the generalibility of the studies findings.
Finally I suggest that authors check again for minor spell check (for ezample line 250: "Additionally" instead of "additionally" and in the same line: "has not been yet" instead of has been not yet".
Author Response
The response has been attached
